# Time–Temperature Superposition Principle in Shearing Tests Compared to Tension Conditions for Polymers Close to Glass Transition

**DOI:** 10.3390/ijms24043944

**Published:** 2023-02-15

**Authors:** Noëlle Billon, Carlos Eloy Federico, Guilhem Rival, Jean Luc Bouvard, Alain Burr

**Affiliations:** Mines Paris, PSL University, Centre for Material Forming (CEMEF), UMR CNRS 7635, 06904 Sophia Antipolis, France

**Keywords:** time–temperature superposition (TTS), shearing, incompressibility, PMMA

## Abstract

The well-known principle of time–temperature superposition (TTS) is of prime interest for polymers close to their glass transition. First demonstrated in the range of linear viscoelasticity, it has been more recently extended to large deformations in tension. However, shear tests were not yet addressed. The present study depicted TTS in shearing conditions and compared it to results in tensile conditions both for low and high strains for a polymethylmethacrylate (PMMA) of different molar masses. The main objectives were to enlighten the relevance of the principle of time–temperature superposition for shearing at high strain and to discuss the way shift factors should be determined. It was suggested that shift factors could be dependent on compressibility, which should be taken into account when addressing various types of complex mechanical loadings.

## 1. Introduction

Modeling and understanding the mechanical behavior of polymers close to their glass transition are still interesting open problems because those materials can be widely used in that domain either for processing amorphous polymers or for end use as semi-crystalline materials.

The glass transition in polymers is not a phase transition. It is a gradual change in the mobility of the polymer chains that are entangled or tangled when the temperature drops. However, this phenomenon is not associated with a critical change in the structure of the polymer. Nevertheless, it is of key importance for polymers because it delimits their fields of application. Studying the behavior of these materials around the glass transition is therefore of great interest. Furthermore, despite many results, some questions remain open.

On the scale of physics, deforming a material means modifying the organization of the atoms that constitute it. The stress defined by the mechanical engineer results from the energy variations involved in these modifications. In the case of polymers, these modifications are of two natures, and their contributions highly depend on the temperature. The interactions between atoms or groups of unbound atoms define the first contribution. The second one is torsion potentials involved in changes in the local conformation of backbone and side groups. When the latter is facilitated by temperature, conformational changes in the backbone become cooperative and introduce a change in the entropy of the system. This is the α transition of the polymer associated with the glass transition.

Eyring [1] proposed that each constituent unit is surrounded by equivalent neighbors that belong to the same chain or not. Each of these units has little room to move within a “virtual cage”. These localized motions are activated by either thermal or mechanical energy or both. The cooperative movements induced by the α transition make the modification of the shape and the volume of each cage possible.

The jump frequency of the change is classically controlled by an activation energy (ψ) necessary for the jump. This frequency is proportional to the term exp−ψkT (where k is the Boltzmann constant and T is the absolute temperature). At equilibrium, there are as many atoms leaving (positive jumps) as atoms entering the cage (negative jumps). The application of a mechanical constraint favors a direction and allows the deformation of the material. This macroscopic deformation is accommodated by many local jumps but depends on the time during which the constraint is applied. Indeed, each jump requires a certain time ($\tau_{\alpha }$) that is proportional to exp+ψkT. If this time is perceptible within the duration of the experience, the behavior will display some time dependence. Consequently, it will be sensitive to the loading rate. On the contrary, if it is very short, the behavior will appear as instantaneous. Finally, if it is very long, the phenomenon will go unnoticed.

This short introduction serves as a reminder of why the macroscopic behavior of polymers is fundamentally viscoelastic. The height of the energy barrier (E) depends on the temperature according to a law for which there is no theoretical form. One generally uses the empirical Vogel, Fulcher, and Tammann (VFT) law [2,3,4], which is given as:(1)$$\tau_{\alpha }=\tau_{0}e^{\frac{A}{T-T_{0}}}$$

This expression can be rewritten as the Williams, Landel, and Ferry (WLF) law [5]:(2) τα=τ0aTTrefaTTref=10−C1T−TrefC2 + T−Tref
where *A*, $T_{0}$, $C_{1}$, and $C_{2}$, are constants. $C_{1}$ and $C_{2}$ depend on the value of the reference temperature $T_{ref}$, which is chosen arbitrarily. aTTref is the so-called shift factor.

Therefore, the behavior of polymers depends on both temperature and time. In the case of a mechanical loading at non-zero, monotonic, or oscillatory velocity, the behavior depends on the temperature and the strain rate ($\dot{\varepsilon }$) or on the temperature and the strain frequency (f).

In addition to the shift factor ($a_{T/T_{ref}}$), a second factor can be introduced ($b_{T/T_{ref}}$) that is often neglected and accounts for the change in density (ρ) due to temperature. The time–temperature superposition principle applies then to the Young’s modulus (*E*) as:(3)Eε˙,Tref=E ε˙ aTTref,T bT/TreforEf,Tref=Ef aTTref,TbT/Tref
where:(4)$$b_{T/T_{ref}}=\frac{\rho \left(T_{ref}\right)\;T_{ref}}{\rho \left(T\right)\;T}$$

Dynamic mechanical analysis (DMA) under forced oscillation, which is one of the key tools used in this analysis, can be used to determine the shift factors as a function of temperature. During this test, the sample is subjected to an oscillatory force or displacement (most often sinusoidal). The complementary quantity (displacement or force) is measured.

In the range in which the behavior remains linear (low deformation), the response has the same form and the same frequency as the stress. For a material with pure elastic behavior, strain and stress remain in phase. If the behavior of the material is viscoelastic, a phase shift (δ) results from the “dissipated” energy at each cycle due to the anelasticity.

Considering the geometry of the loading and assuming that the strain and the stress are uniform, two quantities ($E^{\prime }$ and $E^{\prime \prime }$ in tension or G′ and G″ in shearing) that have the dimension of a modulus can be extracted (Equation (5)):(5)$$\left\{\begin{array}{c} E^{\prime }\;=\;\frac{\sigma_{0}}{\varepsilon_{0}}\cos\delta \;or\;G\prime =\;\frac{\tau_{0}}{\gamma_{0}}\cos\delta \\ E^{\prime \prime }=\;\frac{\sigma_{0}}{\varepsilon_{0}}\sin\delta \;or\;G\prime \prime =\;\frac{\tau_{0}}{\gamma_{0}}\sin\delta \end{array}\right.$$
where $\sigma_{0}$ and $\tau_{0}$ are stress amplitudes and $\varepsilon_{0}$ and $\gamma_{0}$ are strain amplitudes.

$E^{\prime }$ (resp. G′) is the conservation modulus and $E^{\prime \prime }$ (resp. G″) is the loss modulus. The loss angle (δ) is such that (Equation (6)):(6)$$\tan\delta =\frac{E^{\prime \prime }}{E^{\prime }}or\frac{G^{\prime \prime }}{G^{\prime }}$$

The time–temperature superposition is applied to both of these two quantities. As stated previously, this has been established in the deformation range for which the polymer behavior remains linear; i.e., while focusing on low deformations (lower than 0.5% in our case). This is the equivalent of characterizing the double dependence on time and temperature of the elastic modulus or of quantities associated with linear viscoelasticity as described above.

However, TTS can also apply to large deformations (e.g., with a value higher than 1). This was recently illustrated thanks to well-instrumented tensile tests of amorphous [6,7,8] or semi-crystalline polymers [9] and in the case of polymers that exhibited crystallization induced by deformation [10]. It was also highlighted during biaxial loadings encountered during the blowing of bottles [11].

On the other hand, the shear tests were not really addressed with regard to solid polymers (seemingly because tension was much more easily controlled). However, this type of loading path presents a major difference in tension because it is isochoric (i.e., without volume variation). In parallel, the glass transition is clearly influenced not only by the rigidity of the chains but also by the free volume (and therefore the density) of the material. It is thus interesting to investigate the differences between these two modes of solicitation.

Consequently, the main objectives of this study were:Firstly, to enlighten the relevance of the time–temperature superposition principle for shear tests at high strain.Secondly, to discuss the role of volume change due to mechanical loading.

The first of these goals was pursued using shear tests at high strain conducted under well-controlled conditions and dealt with promoting local measurements and constant strain rates in isothermal conditions during the tests. Indeed, despite their widespread use, nominal quantities (referring to the initial dimensions) would not have allowed a rigorous characterization of the state of deformation and stress. In the same manner, very often only the crosshead velocity is kept constant, which implies that the zone of observation would have experienced a variable strain rate. This paper describes the way the tests were conducted to overcome those limitations. The results for two PMMAs of different molar masses will be discussed as a proof of how TTS applies in high-strain shear tests in comparison with tensile conditions. In that part of the study, TTS was calibrated using shear DMA tests.

In a second step, the shift factors (Equation (2)) deduced from shear tests were compared to those obtained in tensile conditions. This was conducted using DMA analyses of amorphous polymers (PMMAs of different molar masses and different glass transition temperatures and a polyetherimide (PEI)) and on a semi-crystalline polyetherketoneketone (PEKK). The effect of the volume change due to mechanical loading was pointed out. We concluded that the shift factors should be revisited to account for that phenomenon.

## 2. Results and Discussion

As stated previously, the database gathered using the above protocol aimed to enlighten the relevance of the time–temperature superposition principle for shear tests (including high strains) and to discuss the role of volume change due to mechanical loading. 

To progress gradually from the first of these conclusions to the last one, we broke down our analyses into three parts:The description of the application of the TTS in shear tests that included a high strain, which was innovative,Its comparison with the tensile loadings (already reported in the past [8]), which also was not found in the current literature. This comparison made it possible to compare conditions with and without variations in volume, and thus to allow us to express the compressibility of the behavior of the material in different ways,The influence of the material’s compressibility on shift factors, which represented a perspective for further improvements.

Altogether, three PMMAs were used that were representative of the wide range of accessible PMMAs. They differed in their glass transition temperatures and in their rubbery behaviors. The α transition temperature ($T_{\alpha }$) did not depend on the type of loading. The $T_{\alpha }$ was defined as the temperature that corresponded to the maximum of the loss angle during temperature scans at a frequency of 1 Hz (Figure 1): 108 °C, 112 °C, and 132 °C for PMMA 80, 93, and 3500, respectively. These differences were an effect of the molar masses. The second effect was the temperature range of the rubbery plateau, which was clearly wider when the molar mass was higher.

The two polymers of extreme molar masses (PMMA 3500 and PMMA 80) made it possible to illustrate that the TTS was applied in the shear as it was applied in the tension [8]. The third (PMMA 93; intermediate mass) reinforced our conclusions regarding the correlation between the incompressibility of the material or the volume change during the loading and the shift factors.

### 2.1. TTS and Shear

Shift factors were estimated for the three materials in the tension and torsion using the Excel^®^ curve-fitting option. To achieve this, frequencies scans were performed every 10 °C. The WLF parameters that were identified for a reference temperature of 130 °C are gathered in Table 1. In that table, the “S” superscript refers to shear measurements, whereas the “T” superscript refers to tensile ones.

Using the shift factors extracted from the torsion DMA tests (aTTrefS), it was possible to demonstrate that the TTS could be applied to the shear at high strain like it was in the tension [6,7,8,9,10,11] and regardless of the PMMA (Figure 2). In other words, the mechanical behavior under shear could be referred to using an equivalent strain rate at a reference temperature (ε˙TrefS= ε˙aTTrefS) to combine the effects of temperature and time.

To go further, as suggested in [12], it can be beneficial to use the α transition temperature ($T_{\alpha }$) as reference temperature (108 °C for PMMA 80 and 132 °C for PMMA 3500) to compare materials. The equivalent strain rates at 130 °C and at $T_{\alpha }$ were then gathered (Table 2). Figure 3 illustrates this latter point by comparing the shear behavior of PMMA 3500 to that of PMMA 80 for ${\dot{\varepsilon }}_{eq_{\alpha }}$ that remained close two by two (Table 3). 

It could be observed that as in tension, the nature of the behavior under shear (elastoplastic, viscoelastic, or hyperelastic) was related to the equivalent strain rate at $T_{\alpha }$ and ${\dot{\varepsilon }}_{eq_{\alpha }}$ regardless of the PMMA considered. This is of practical interest in efficiently comparing materials that erase the effect of the difference in the glass transition temperature. So, we observed in the last case (the lower strain rate) that PMMA 80 was close to its flowing zone, whereas PMMA 3500 was still hyperelastic due to its molar mass (see (3) in Figure 3).

### 2.2. Comparison between Tension and Shear

For a given equivalent strain rate at the same reference temperature, the type of macroscopic behavior (elastoplastic, viscoelastic, or hyperelastic) was the same in the tension and in the shearing (Figure 4). 

However, Table 2 shows a small difference between the equivalent strain rates deduced from shearing and those deduced from tension. Indeed, aTTrefS factors deduced from torsion and those extracted from tensile tests (aTTrefT) were different (Figure 5 and Figure 6).

Even if the differences appear small, two sets of experiments for PMMA 93 using two injection-molding campaigns and two sets of machines were conducted to highlight the order of magnitude of the differences between the shearing and tension. 

In addition, it can be seen in Figure 5 that the differences could be up to 15% (although they appeared small on log-scales) depending on the material and on the temperature, which was only 5 degrees apart from the reference temperature. This of course could have had an impact on modeling based on the notion of the equivalent strain rate as a parameter in complex 3D loadings [13,14]. The questions then regarded how to estimate the shift factors and whether should they depend on some parameters related to the type of loading (shear, tension, or compression).

The same general trends were observed for other polymers such as PEI and PEKK (Figure 7 and Table 1), but the amplitude of the differences depended on the polymer. 

In the case of amorphous polymers, aTTrefS always appeared to be lower than or equal to aTTrefT (Figure 6). From a practical point of view, this implied that reaching the same equivalent strain rate at a reference temperature could be achieved with a higher strain rate in the tension than in the torsion. This could not be anticipated only by examining master curves (Figure 8). 

From a more physical point of view, the same effect would have been obtained if a density correction (Equation (3)) had been applied to account for a higher density in the torsion than in the tension. It was then tempting to relate this to the free volume that could be increased in the tension while it remained constant under shearing, which is known to be an isochoric phenomenon.

### 2.3. TTS and Incompressibility

One additional clue was the PMMA 93 that exhibited a tensile behavior close to incompressibility between an equivalent strain rate of 0.0001 to 5 s^−1^ at 130 °C up to a strain of 0.5. This was validated thanks to measurements of the volume variation during tension as described above.

To limit the range of the strain rate for which the polymer would be incompressible in the conditions in which we identified the shift factors, we analyzed the master curves. The low deformation simplified the analysis, and we could assume that an apparent Poisson’s ratio could be defined. If the behavior of a material is linear elastic, it should exhibit a Poisson coefficient (υ) of 0.5, and the tensile modulus I would be related to the shear modulus (G) through Equation (7):(7)$$G\;=\;\frac{E}{2\left(1+\upsilon \right)}=\;\frac{E}{3}$$

In the case of linear viscoelasticity, the apparent modulus should be close to the norm of the complex moduli ($E^{\prime }+iE^{\prime \prime }or\;G^{\prime }+i\;G\prime \prime )$; i.e.:(8)$$\left\{\begin{array}{c} E^{*}=\sqrt{{E^{\prime }}^{2}+{E^{\prime \prime }}^{2}}\\ G^{*}=\sqrt{{G^{\prime }}^{2}+{G^{\prime \prime }}^{2}}=\frac{E^{*}}{2\left(1+\nu \right)} \end{array}\right.$$

In such a case, one can assume that:(9)$$\left\{\begin{array}{c} G^{\prime }\;=\;\frac{E^{\prime }}{3}\\ G^{\prime \prime }\;=\;\frac{E^{\prime \prime }}{3} \end{array}\right.$$

It was possible to validate this observation in the DMA conditions shown in Figure 8, in which the tensile (*E′* and *E″*) and torsion (*G′* and *G″*) master curves at 130 °C for PMMA 93 are plotted. The estimations of *G′* and *G″* from *E′* and *E″* and Equation (9) is superimposed in grey.

At this stage, we could assume that the behavior of PMMA 93 was close to incompressibility in the range of the equivalent strain rate for which the polymer exhibited a viscoelastic or rubbery behavior (i.e., from 10^−4^ to 1 s^−1^ at 130 °C). Due to TTS itself, this corresponded to the range of temperatures “surrounding” the α transition at 1 s^−1^; i.e., 112 °C for PMMA 93 and 132 °C for PMMA 3500.

Figure 6 shows that this temperature corresponded to the zone in which aTTrefS and aTTrefT were almost equal.

This suggested that there could have a correlation between the difference between the two factors and the incompressibility of the behavior. However, such an observation would have to be confirmed.

## 3. Materials and Protocols

### 3.1. Materials

PMMAs of three different molar masses supplied by Arkema Company were used. One of high mass (PMMA 3500, Mw of 3500 kg/mol, Mn of 880 kg/mol) was provided as casted 4 mm-thick plates. The others of lower mass (PMMA 93, Mw of 93 kg/mol, Mn of 44 kg/mol and PMMA 80, Mw of 80 kg/mol, Mn of 42 kg/mol) were provided as 4 mm-thick plates that were injection molded. The dimensions of the plates were 300 × 300 mm and 100 × 100 mm, respectively. For the following analysis, the samples were processed at various dimensions using a 3-axis cutting machine.

### 3.2. DMA Analysis

The tensile DMA tests were performed using 30 mm × 4 mm × 1 mm rectangular samples. The effective gauge length was 5 mm. The maximum strain was 0.001 and the heating rate was 1 °C/min for temperature-scanning tests. From a technical point of view, it was difficult to reach a flowing zone in tension. Conversely, by combining oscillatory rheology and torsion DMA, a wider range of temperatures was explored in shearing. Shear loading in the solid state resulted from torsion of the rectangular samples (30 mm × 4 mm × 1 mm), while the parallel discs were used for the flowing state (discs with a diameter of 25 mm and a 1 mm thickness).

For comparison, some results for ULTEM 1010^®^ PEI and KEPSTAN 7002^®^PEKK from Arkema (Colombes, France) were gathered on 2 mm-thick injection-molded plates. The tensile samples were then 20 mm × 4 mm × 2 mm, whereas the torsion ones were 35 mm × 10 mm × 2 mm. The maximum strain was 0.004.

The different geometries are depicted in Figure 9a. The master curves were built from isothermal steps for frequencies ranging from 0.1 Hz to 100 Hz every 10 °C. This was performed manually using the same protocol as previous studies [8]. 

### 3.3. High Strain Tests

High-strain tests consisted of tensile tests aimed at exploring compressibility behavior and shearing tests aimed at exploring TTS in the latter conditions. In both cases, digital image correlation (DIC) was used to assess local strains and stresses.

Since all tests were performed above the glass transition temperature, the materials were considered as renewed by the pre-heating (10 min). Experiments were then performed without any additional pre-conditioning.

**(a)** 
**Tensile tests**


Dumbbell-shaped samples were tooled from the 4 mm-thick PMMA sheets (Figure 9b) in the perpendicular direction to the injection flow. Samples were designed with a 10 mm-long straight-walled section at the center to localize strain while avoiding triaxial effects in the measurement area. The tensile tests were performed using an exponential crosshead velocity. As a result, the material was strained at constant strain rates in the central area and in uniaxial conditions. 

A random pattern was sprayed on the surfaces (face and thickness) of the samples. A high-sampling-rate video camera was used to record the tests. The sample thickness was recorded as well to assess the strain in the third (thickness) direction. This was made possible by using a prism that reflected the image of the thickness to the front camera (Figure 9b).

The strain field was assessed via the deformation of the pattern using image-correlation software. Hence, the local strains in the 3 main directions were known. Hencky’s strains were used. Finally, the volumetric strain during the test was calculated using the following equation [12]:(10)$$\frac{\Delta V}{V_{o}}=exp\left(\varepsilon_{xx}+\varepsilon_{yy}+\varepsilon_{zz}\right)-1$$
where $\varepsilon_{zz}$, $\varepsilon_{yy}$, and $\varepsilon_{xx}$ are the true strain in the thickness, transverse, and longitudinal directions, respectively; and $V_{o}$ is the initial volume.

**(b)** 
**Shearing tests**


An Iosipescu configuration [15] with a specific angle of 90° to promote a local shear condition [16] was used. The sample geometry and the experimental set-up are displayed in Figure 9c and Figure 10a, respectively.

Similar to tensile loading tests, an exponential crosshead velocity was imposed to promote a local true strain rate as constant as possible. DIC allowed measurement of the true strains on the front face.

Prior using this set-up, a validation was conducted to assure that the local strain was mainly under shear conditions, and a value of 25% was chosen as a maximum strain to guarantee “pure” local shear with no sliding. We observed that the deformation was quasi-homogeneous in the central zone (Figure 9c). However, when examining it in detail, it varied close to the notch. This illustrated the interest of local measurements.

According to [16], the mechanical response under shear loading can be characterized thanks to the apparent shear stress (Equation (11)):(11)$$\sigma_{xy}=\frac{F}{AB\cdot h}$$
where *AB* is the length between the notches and *h* is the thickness (refer to Figure 9c). 

However, the stress and strain were not perfectly uniform along the ligament due to the presence of notches that induced stress concentration. So, we had to define a strategy to address the local true strain and true stress at one given point. Thus, a “zone of interest” was chosen (square in Figure 9c) at the center of the midsection of the ligament. The strain was deduced from the digital image correlation (DIC) facility. As for the stress, it had to be estimated from $\sigma_{xy}\;$(Equation (11)), which was the only quantity accessible to experience. As suggested by several authors in the past, we applied a correcting factor ($C_{s}$) to the measured stress to account for this non-uniformity [17,18,19]. Following these authors, $C_{s}$ is the average ratio between the shear stress at a given point in the midsection of the sample ($\tau_{xy}$) and the average shear stress in the midsection ($❬\tau_{xy}❭$) (Equation (12)). These two values were estimated using finite element calculations carried out in ABAQUS 6.14-1. Figure 10b displays the mesh (consisting of a 10-node quadratic 1.5 mm tetrahedron with hybrid formulation and constant pressure elements (ABAQUS C3D10H)). The left side was fixed in all three directions of the space, while an exponential displacement was applied to the right side of the sample. 

The initial approach was carried out within the frame of elasticity. To account for both potential sensitivity to the behavior and the effect of the drastic change in behavior close to glass transition, two simulations were performed: the first one assumed a hyperelastic behavior [20,21] of the PMMA (rubbery regime), and the second one accounted for elastoplastic behavior (glassy state). Both were fitted on experimental data and allowed to reproduce experimental stress vs. strain curves up to a strain of 0.25. Figure 10b shows one typical stress contour and illustrates the point at which $\tau_{xy}$ was gathered and the section on which stress was averaged to $❬\tau_{xy}❭$. Then:(12)$$C_{s}=\frac{\tau_{xy}}{❬\tau_{xy}❭}$$

For hyperelastic material, the stress correction factor was equal to 1.091, which was in agreement with previous estimates [16,17,18,19]. The same parameter was used in the viscoelastic regime. For the elastoplastic behavior (glassy region), a factor of 1.342 was used.

Finally, the corrected average shear stress ($\sigma^{c}{}_{xy}$), which was closer to the local stress (where the strain was measured) was calculated using Equation (13):(13)$$\sigma^{c}{}_{xy}=\frac{F}{C_{s}\cdot AB\cdot h}$$

## 4. Conclusions

The mechanical behavior under shear of PMMAs of different molecular weights was investigated at low strain using DMA from the solid to the quasi-liquid state and at high deformation by performing loading–unloading tests with DIC techniques and an Iosipescu apparatus.

The strain rate/temperature superposition principle was used to build the storage modulus and loss modulus master curves from low-strain measurements. Then, shift factors were determined and fitted using the Williams–Landel–Ferry equation.

Using these shift factors, equivalent strain rates at the reference temperature could be defined for any sets of coupled temperature–strain rate (including experimental conditions at high strain). The same equivalent strain rate led to the same type of behavior (elastoplastic, viscoelastic, or hyperelastic) in the tension and under shearing and to the same behavior for each of the given loading geometries. The use of equivalent strain rates at α transition temperature allowed us to define a type of universal scale of strain rates for comparing different materials.

Complementary to previous studies performed under tension, one of the main conclusions of this paper was that the time–temperature superposition principle applied in the region of the glass transition of polymers regardless of the loading (even at high strain). This suggested that kinetics of elementary processes are equivalent at any level of strain and are only based upon small changes in local conformation. Differences between high and low strains would then be due to the number of active elementary processes that are accumulated during the history of loading.

However, differences existed between shift factors extracted from tension and those extracted from torsion tests. This could be observed for various polymers in addition to PMMA.

These differences could be due to the fact that the shearing was a loading condition that kept the volume unchanged, whereas the volume could increase during tension. It was then suggested that shift factors could be sensitive to the interrelation between volume changes due to mechanical loading and the free volume in the polymer.

This also suggested that they should be corrected to account for compressibility in the behavior, which can depend on the temperature and strain rate.

A solution could be to reintroduce the density correction (Equation (3)) and to express the dependence of the density not only on the temperature but also on the deformation. The first invariant or volume variation would be the parameter to use.

One could then define a usual shift (e.g., a WLF or Arrhenius-like) factor and a second shift depending on compressibility of the behavior in an manner inspired by the density correction (Equations (3) and (4)).

Obviously, additional studies should be performed to validate this idea using various conditions of loading: compression, biaxial, etc.

## Figures and Tables

**Figure 1 ijms-24-03944-f001:**
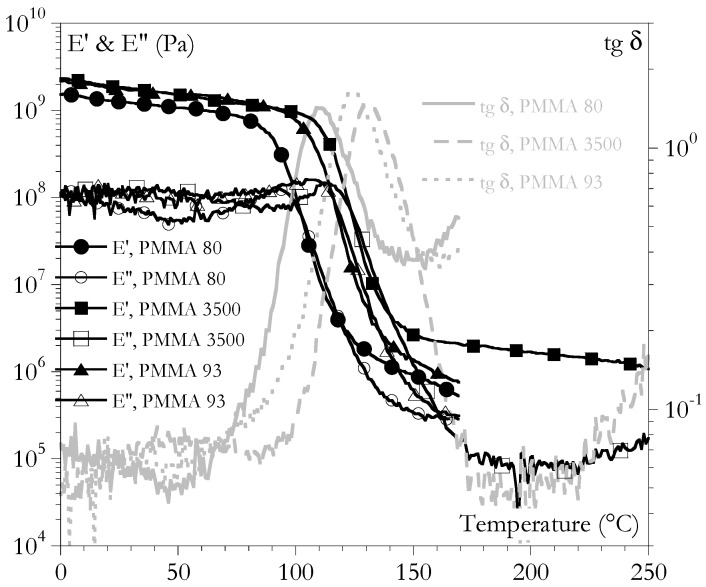
Temperature scans for the three PMMAs at a heating rate of 1 °/min and a frequency of 1 Hz.

**Figure 2 ijms-24-03944-f002:**
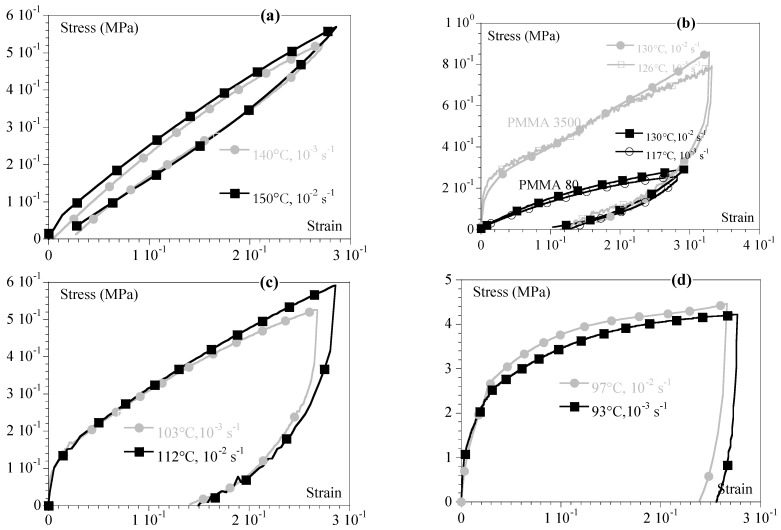
Shear behavior of PMMA 80 and PMMA 3500 for different sets of conditions (given in the legends) as a function of equivalent strain rate at 130 °C (Table 2): (**a**) PMMA 3500—${\dot{\varepsilon }}_{eq_{130}}≅10^{-4}\;s^{-1}$; (**b**) PMMA 3500—${\dot{\varepsilon }}_{eq_{130}}≅6\;\times \;10^{-3}\;s^{-1}$ and PMMA 80—${\dot{\varepsilon }}_{eq_{130}}≅10^{-2}\;s^{-1}$; (**c**) PMMA 80—${\dot{\varepsilon }}_{eq_{130}}≅4\;\times \;10^{-3}\;s^{-1}$; (**d**) PMMA 80—${\dot{\varepsilon }}_{eq_{130}}≅0.5\;s^{-1}$.

**Figure 3 ijms-24-03944-f003:**
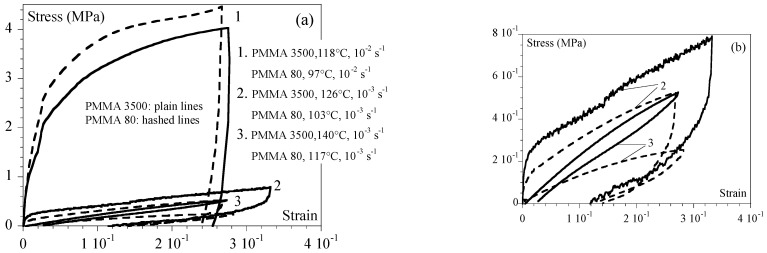
Comparison between PMMA 3500 (plain lines) and PMMA 80 (hashed lines) for similar equivalent strain rates at T_α: (1) PMMA 3500—118 °C and 0.01 s^−1^ and PMMA 80—97 °C and 0.01 s^−1^; (2) PMMA 3500—126 °C and 0.001 s^−1^ and PMMA 80—103 °C and 0.001 s^−1^; (3) PMMA 3500—140 °C and 0.001 s^−1^ and PMMA 80—117 °C and 0.001 s^−1.^ Equivalent strain rates are given in Table 2. (**b**) Zoom of (**a**) for better visibility.

**Figure 4 ijms-24-03944-f004:**
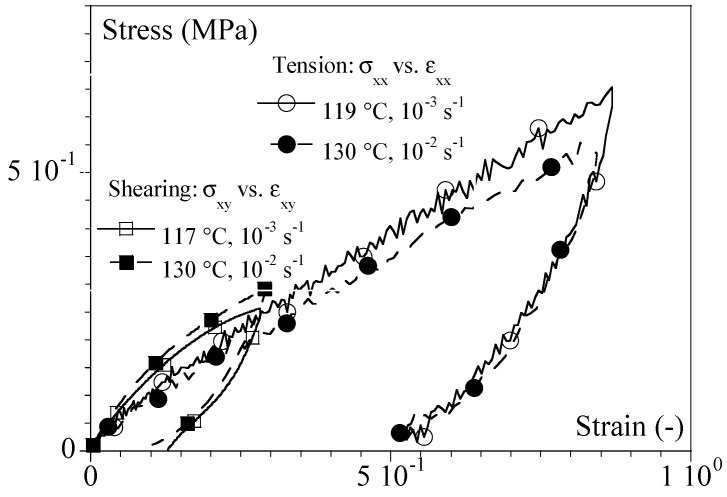
Comparison between tension and shearing for PMMA 80 at an equivalent strain rate of 10^−2^ s^−1^ at 130 °C. Stress (*σ*) vs. strain (*ε*) curves for two equivalent experimental sets of conditions are given in the legend. Squares represent shearing ($\sigma_{xy}\;vs.\;\varepsilon_{xy}$); circles represent tension ($\sigma_{xx}\;vs.\;\varepsilon_{xx}$ ).

**Figure 5 ijms-24-03944-f005:**
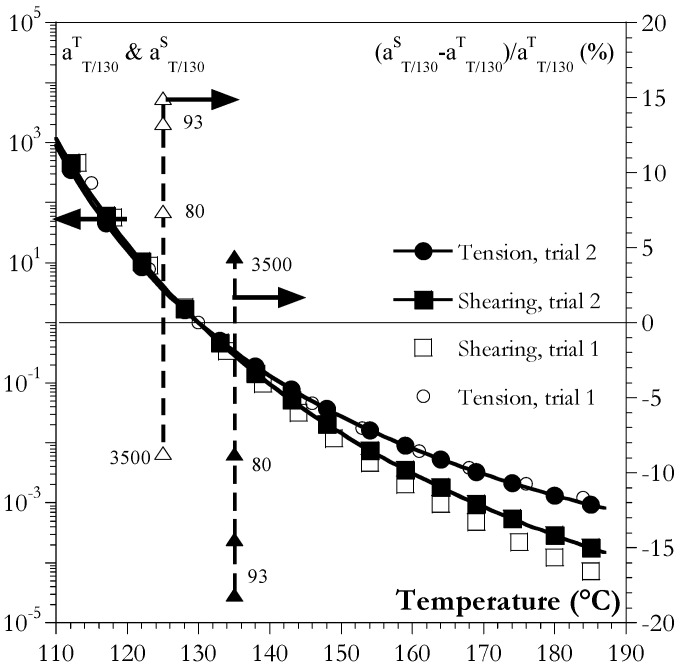
Comparison between aTTrefS and, aTTrefT (scale on the left side) for PMMA 93 using two sets of trials (plain and hollow squares and circles). Triangles (scale on the right side) depict the relative differences in shift factors between the tension and torsion in % for two arbitrary temperatures (resp. $T_{ref}-5\;{}^{\circ}C$ and $T_{ref}+5\;{}^{\circ}C$ ) and for all PMMA (figures refer to their identifiers). Curves were recalculated from parameters given in Table 1.

**Figure 6 ijms-24-03944-f006:**
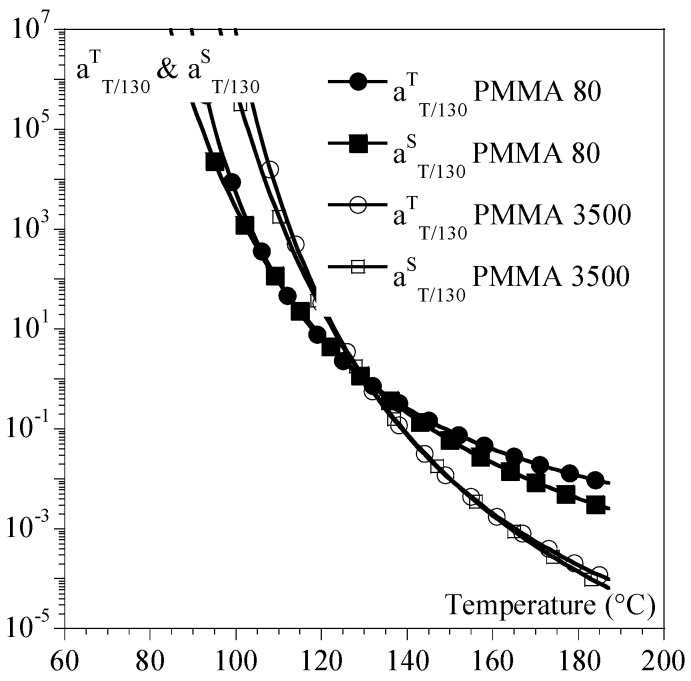
Comparison between aTTrefS (circles) and aTTrefT (squares) for PMMA 80 (plain symbols) and PMMA 3500 (hollow symbols). Curves were recalculated from parameters given in Table 1.

**Figure 7 ijms-24-03944-f007:**
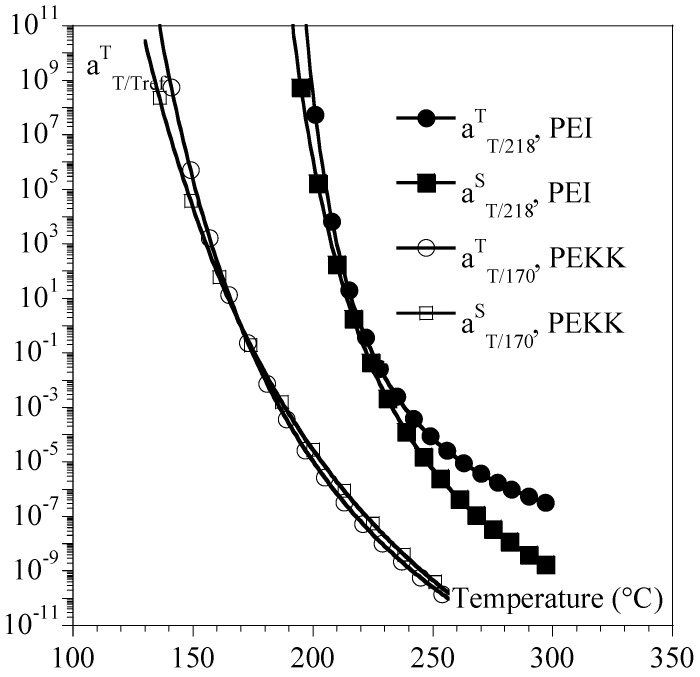
Comparison between aTTrefS (circles) and aTTrefT (squares) for PEI (plain symbols) and PEKK (hollow symbols). Curves were recalculated from parameters in Table 1.

**Figure 8 ijms-24-03944-f008:**
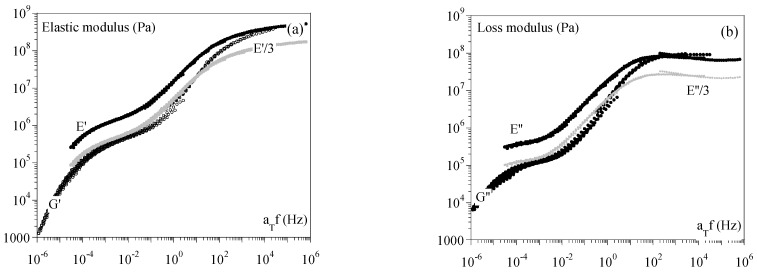
Master curves at 130 °C for PMMA 93. Comparison between torsion (G′ and G″) and tension (E′ and E″). Left: elastic moduli; right: loss moduli. Grey curves correspond to estimates of G′ (resp. G″) from E′ (resp. E″) while assuming incompressibility (υ=0.5). (**a**) Elastic modulus; (**b**) Loss modulus.

**Figure 9 ijms-24-03944-f009:**
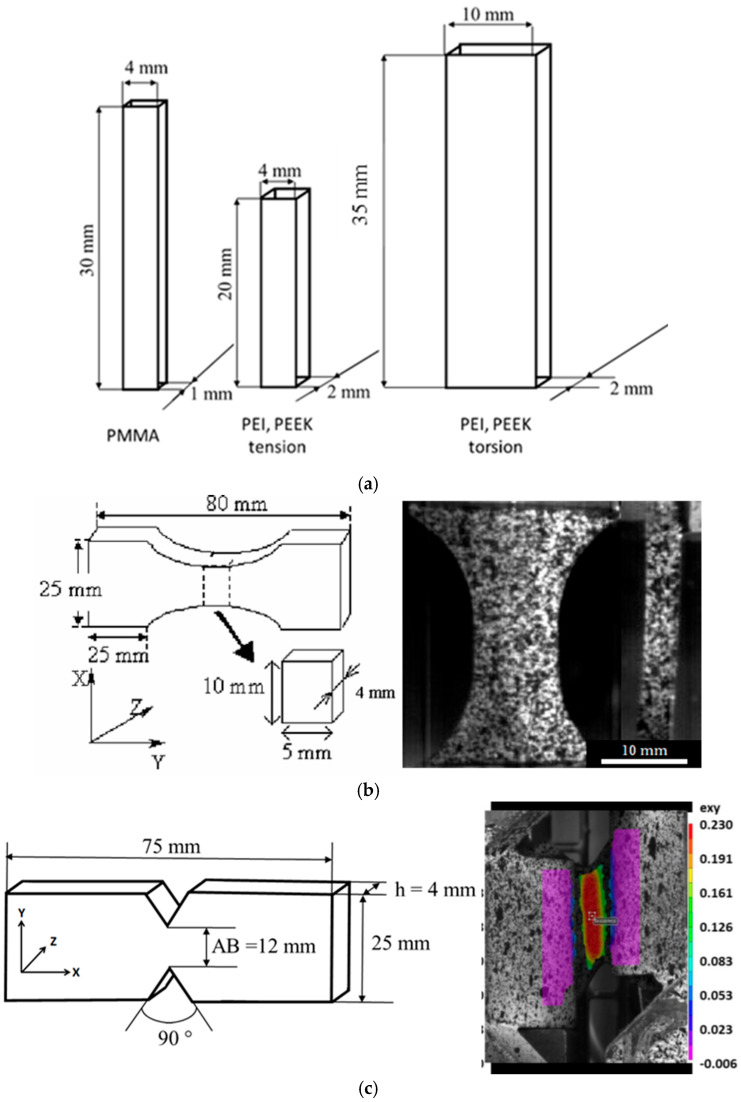
Samples of geometries and dimensions: (**a**) DMA samples; (**b**) tensile specimen and image showing the random pattern (face and the thickness); (**c**) shear sample and strain (the square represents the zone of interest for which the stress and strain were estimated).

**Figure 10 ijms-24-03944-f010:**
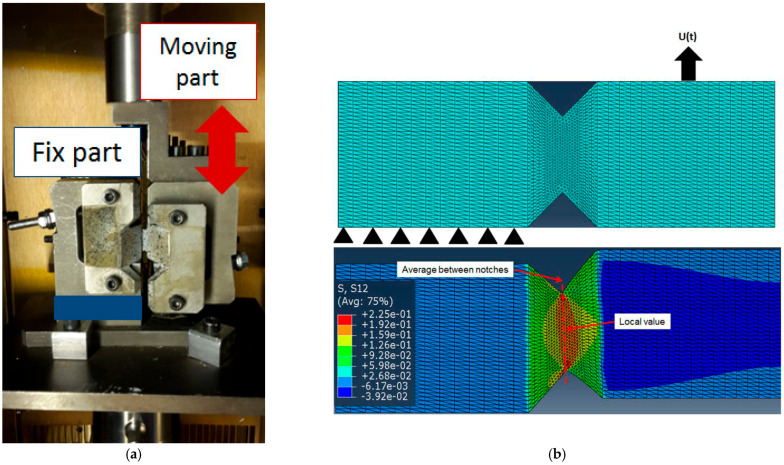
(**a**) Iosipescu configuration. (**b**) Mesh and boundary conditions for simulations (top right) and equiv-alent stress contour of the deformed hyperelastic specimen (bottom right).

**Table 1 ijms-24-03944-t001:** WLF parameters (Equation (2)) for the three PMMAs, PEI, and PEKK. The “T” exponent refers to measurements of tension, whereas “S” refers to measurements of torsion.

PMMA	Tension		Shear		$T_{ref}$ (°C)
	$C_{1}^{T}$	$C_{2}^{T}$	$C_{1}^{S}$	$C_{1}^{S}$	
80	4.5	66.3	6.6	87.9	130
93 (Trial 1)	5.7	52	10.5	84	130
93 (Trial 2)	6.6	64.7	9.2	80	130
3500	8.8	68.1	10.3	83.3	130
PEI	10.2	44	16	65	218
PEKK	21.7	91	25.3	137	170

**Table 2 ijms-24-03944-t002:** Experimental conditions for shear tests and corresponding equivalent strain rates at 130 and at $T_{\alpha }$ in shearing and tension.

9		Experimental Conditions		Equivalent Strain Rates		
PMMA	$T_{\alpha }$ $\left({}^{\circ}C\right)$	$T_{exp}$ $\left({}^{\circ}C\right)$	$\dot{\varepsilon }$ $\left(s^{-1}\right)$	ε ˙aT130S $\left(s^{-1}\right)$	ε ˙aTTαS $\left(s^{-1}\right)$	ε ˙aT130T $\left(s^{-1}\right)$
80	108	117	$10^{-3}$	$1.4\times 10^{-2}$	$8.8\times 10^{-5}$	$1.3\times 10^{-2}$
80	108	130	$10^{-2}$	$10^{-2}$	$6.3\times 10^{-5}$	$10^{-2}$
80	108	103	$10^{-3}$	0.84	$5.3\times 10^{-3}$	1.2
80	108	112	$10^{-2}$	0.5	$3.1\times 10^{-3}$	0.48
80	108	93	$10^{-3}$	63	0.39	481
80	108	97	$10^{-2}$	93	0.58	289
3500	132	140	$10^{-3}$	$7.9\times 10^{-5}$	$1.4\times 10^{-4}$	$7.5\times 10^{-5}$
3500	132	150	$10^{-2}$	$10^{-4}$	$1.8\times 10^{-4}$	$10^{-4}$
3500	132	126	$10^{-3}$	$3.3\times 10^{-3}$	$5.8\times 10^{-3}$	$3.5\times 10^{-3}$
3500	132	130	$10^{-2}$	$10^{-2}$	$1.7\times 10^{-2}$	$10^{-2}$
3500	132	118	$10^{-2}$	0.54	0.94	0.76

**Table 3 ijms-24-03944-t003:** Comparison of PMMA 3500 and PMMA 80 in equivalent states. Experimental conditions as used in Figure 3 and with the associated equivalent strain rate at $T_{\alpha }$.

Polymer	Strain Rate (s^−1^)	Temperature (°C)	${\dot{\varepsilon }}_{eq_{\alpha }}(s^{-1})$
PMMA 3500	0.01	118	0.94
PMMA 80	0.01	97	0.58
PMMA 3500	0.001	126	$5.8\times 10^{-3}$
PMMA 80	0.001	103	$5.3\times 10^{-3}$
PMMA 3500	0.001	140	$1.4\times 10^{-4}$
PMMA 80	0.001	117	$8.8\times 10^{-5}$

## Data Availability

No additional data.

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
