# Peer review of "Time–Temperature Superposition Principle in Shearing Tests Compared to Tension Conditions for Polymers Close to Glass Transition"

_ijms, 2023, doi:10.3390/ijms24043944_

Round 1
Reviewer 1 Report
· Page 4: Since the Young’s modulus and the shear modulus are related by the Poisson’s coefficient, eq. 6 is somehow equivalent to assume that the Poisson’s coefficient is independent of time and temperature. This does not hold for all the kind of polymers. This assumption should be discussed and justified.
· Figures of the specimens (for all the considered materials), as well as a scheme of the experimental setup used for tensile and shear tests should be added to sect. I.2.
· Page 7, formulas 8, 9, and 10. The authors denote by \sigma_{xy} the mean shear stress on the considered section. Then, they state that the maximum stress is given by the mean stress, incremented by a factor C_s>1, that is numerically evaluated. It is not clear how they (eq. 10) evaluate a “corrected” shear stress by dividing (instead of multiplying) the mean stress by C_s.
· Page 8. The corrective factor C_s is evaluated numerically. It should be better explained how the materials are modelled, also recording the material parameters. Is the viscoelasticity accounted for? What test rate has been used in the simulation. For the elastoplastic material, is the elastic limit overcome? Furthermore, since the considered materials are nonlinear, one should expect a value of C_s varying during the test (i.e., different values of C_s for different load levels).
· The presentation of the results (Sect. II) is quite confuse and it should be strongly improved.
· Results found at page 15 confirm that the Poisson’s ratio is time- (or frequency-) dependent. Since this provides the different response in tension and shear (since this influences the ratio between E and G), this conclusion should be better and deeply discussed.
· 10/20 of the references are self-citations!
Minor comments:
· Letters representing physical quantities/variables should be italicized in the text
· Punctuation after equations is missed
· Page 2, before eq. 1: add reference for the VFT law
· Uppercase E is used for both the energy and the Young’s modulus. This could be misleading for the reader
· Notation used in eq. 3 is unclear
· Page 3, after eq. 3: it is not clear what “horizontal” and “vertical” are referred to. Maybe a graph could be added for clearness
· Eq. 5: in structural mechanics, shear strain and stress are usually denoted to ad \gamma and \tau, respectively. The definition of the shear modulus should follow this notation.
· Page 4: The sentence “Yet, this type of loading presents a major difference with tension as it is isochoric” should be rephrased (for example by specifying that shear deformations entail no volume variation). Indeed, the term “isochoric” is rarely used in solid mechanic; it’s mainly used in thermodynamics.
· Page 4: the use of commercial brands should be avoided in scientific papers
· Eq. 8: A, B, h and h’ should be highlighted in fig. 2.
· After eq.8: the difference between h and h’ should be clarified.
· Page 7: to use both \sigma and \tau to denote the shear stress could be misleading for the reader.
· Page 8: the figure number is missed.
· Figure 3 could be improved by adding plots of the loss angle for the three materials
· Excel is misspelt
· Page 10, line 3: “gathered it” should be changed into “gathered in”
· Fig.s 5 and 6: legend should be added to the graphs
Author Response
Thank you very much for the time and valuable remarks that will help improving the manuscript.
We answered all the request and corrected the text to make it more understandable to anybody. Please see the attachment
According to other review some amendments were done and appears yellow underlined.
A specific effort was made to improve the English.
The precise answers (in blue) follow reminding the requests (in black) are given in the attachment

Reviewer 2 Report
This paper is reporting the rheological performance of PMMA, especially focusing on the time-temperature superposition (tTs) principle on shear and elongational modes. I believe that the manuscript needs considerable reworking in order to deliver its message. The conclusions are very weak and not in focus (as mentioned below, the selection of polymers and data presented are not coherent), while it is not fully clear to me the difference to earlier works on this well-known material and method.
Below some further comments in order of appearance; importance varies.
Abstract:
"the shear tests" --> "shear tests"
"Present paper" --> "The present paper"
"Main objectives" --> "The main objectives"
The abbreviations PMMA, PEI, and PEKK need to be spelled out at their first appearance (also in the abstract). DIC is spelled out twice; can be reduced.
Page 2: "...of two natures of importance...". Please rephrase for better syntax?
Page 2 (2x): "...of the skeleton...". I think "backbone" would be a more accurate description of the polymeric chain here.
Page 4: "Paper depicts" --> "The paper depicts"
Page 4: What were the other two dimensions of the PMMA plaques?
Page 5: I assume the DMA specimens were obtained from the above mentioned plaques of PMMA? How was the final thickness of 1 mm reached? And what about the 4 mm tensile tests? How did they result from the 3 mm thick plaques?
Page 5, DMA analysis: Please use a space between the value and the unit (e.g. 30mm --> 30 mm).
Page 5: "...above glass transition temperature..." --> "above the glass transition temperature..."
Page 5: "...reflecting the image of the thickness". I think the syntax needs to be improved to make sense. Probably the thickness of the sample is meant?
Figure 1b: A scale is missing.
Figure 2 caption: Issue with reference.
Page 9: "Shift factors was estimated..." --> "Shift factors were estimated..."
Page 9: Excell --> Excel
Page 10: "...as suggested in a recent paper [14]...". Reference 14 is from 2002, already 20 years old. I would't call it recent any more?
Page 10: "...and the figure 5..." --> "...and Figure 5..."
Page 10: The results of the buller points would be better readable in a tabulated form.
Page 10: "...erasing effect..." --> "...erasing the effect..."
In Figure 4, 5, 6, 8 and Table 2 no results of the sample PMMA 93 are shown. On the other hand, in Figure 7, 10 and 11 (sample missing in caption?) it is only PMMA 93 present, and not the other two samples. This needs to be supplemented and/or explained why this choice was made?
Page 11: "As differences...of scattering". Please rephrase to improve syntax.
Page 12: "Same general trends..." --> "The same general trends..."
Page 12: "...but amplitude..." --> "...but the amplitude..."
Author Response

(The authors gave the same response as above.)

Round 2
Reviewer 1 Report
The authors have updated the paper by following the reviewer’s comments, so improving its quality. In particular, the new version of Fig.s 1 and 2 are very clear. However, there are still areas that should be improved before publication:
· I understand that the authors are considering the Poisson’s ratio as defined in linear elasticity. However, for a wide class of polymer the Poisson’s ratio is dependent on time and temperature: see, among the others, [Yang, L., Yang, L., & Lowe, R. L. (2021). A viscoelasticity model for polymers: Time, temperature, and hydrostatic pressure dependent Young's modulus and Poisson's ratio across transition temperatures and pressures. Mechanics of Materials, 157, 103839.], [Pandini, S., & Pegoretti, A. (2011). Time and temperature effects on Poisson's ratio of poly (butylene terephthalate). Express Polymer Letters, 5(8).], [Pandini, S., & Pegoretti, A. (2008). Time, temperature, and strain effects on viscoelastic Poisson's ratio of epoxy resins. Polymer Engineering & Science, 48(7), 1434-1441.]. Surely, one can perform an approximated analysis by assuming that the Poisson’s ratio is time- and temperature- independent, but this should be clearly stated.
· It’s still not clear why, if the corrective factor C_s is defined as the maximum stress divided by the mean stress (eq. 9), and hence is >1, the “corrected stress” (eq.10, that seems to be the maximum stress to which the material is subjected) is evaluated by DIVIDING the mean stress by the correction factor (instead of multiplying). In such a way, the maximum stress turns out to be LOWER than the mean stress. This point should be clarified (also because the cited references are very dated, and difficult to find online).
· Evaluation of the corrective coefficient: the authors have improved the description of the numerical simulation performed to evaluate C_s. However, the (possible) dependence of this coefficient on the load level is still unclear. Indeed, since the considered materials are nonlinear, one should expect a value of C_s varying during the test (also depending on the possible overcoming of the elastic limit, for the elastoplastic material). Furthermore, it seems that viscoelasticity is not accounted for in the simulations. Also this point should be discussed. From the new description of the evaluation of C_s, it seems that the value correspondent to a strain of 0.25 has been considered.
Minor comments:
· Letters representing physical quantities/variables should be italicized in the text
· Eq. 3.b contains a misprint
· Before eq. 7: ref [124], not appearing in the reference list, is cited.
· The sentence “If the behavior of the material were linear elastic, it should exhibit a Poisson coefficient, ?, of 0.5” should be misleading, since the value of 0.5 for the Poisson’s ratio is related to the incompressibility of the material, not to its linear elasticity.
Author Response
Thank you for valuable remarks.
We have accounted for each of them.
Please see the attachment.

Reviewer 2 Report
The authors addressed all points, significantly improving the manuscript., thus publication is supported.
English check of the newly added text would be needed in the publishing process.
Author Response
Thank you for kind review
See attachment

Round 3
Reviewer 1 Report
the authors revised the paper in a satisfying way